# Decrease in Bone Formation and Bone Resorption during Intravenous Methylprednisolone Pulse Therapy in Patients with Graves’ Orbitopathy

**DOI:** 10.3390/jcm11175005

**Published:** 2022-08-26

**Authors:** Joanna Rymuza, Klaudia Gutowska, Dagmara Kurpios-Piec, Marta Struga, Piotr Miśkiewicz

**Affiliations:** 1Department of Internal Medicine and Endocrinology, Medical University of Warsaw, Banacha 1a, 02-097 Warsaw, Poland; 2Doctoral School, Medical University of Warsaw, Zwirki i Wigury 81, 02-091 Warsaw, Poland; 3Department of Biochemistry, Medical University of Warsaw, Banacha 1a, 02-097 Warsaw, Poland

**Keywords:** bone turnover marker, intravenous methylprednisolone, intravenous glucocorticoids, Graves’ orbitopathy, Graves’ ophthalmopathy

## Abstract

Background: Treatment with glucocorticoids (GCs) is associated with side effects. In contrast to the well-known negative impact on bone tissue exerted by oral GCs, few data are available regarding intravenous GCs. We investigated the influence of intravenous methylprednisolone (IVMP) on bone turnover markers (BTM): amino-terminal propeptide of type I procollagen (P1NP) and the C-terminal telopeptide of type I collagen (CTX), and on calcium metabolism parameters: 1,25-dihydroxyvitamin D (1,25(OH)_2_D), 25-hydroxyvitamin D (25(OH)D), calcium (Ca), phosphate (P), and intact parathormone (iPTH). Methods: In a prospective study, 23 consecutive subjects with Graves’ orbitopathy were included and treated with IVMP according to the European Group on Graves’ Orbitopathy recommendations. We evaluated effects on BTM occurring during the first 7 days after 0.5 g IVMP, and after the therapy with 12 IVMP pulses with a cumulative dose of 4.5 g. Results: We observed prompt but transient decrease of P1NP (*p* < 0.001) and the reduction of CTX (*p* = 0.02) after the first IVMP pulse. Following the full course of IVMP therapy, both P1NP and CTX were found decreased (*p* < 0.05 and *p* < 0.01, respectively). Conclusions: A single pulse of 0.5 g IVMP already decreases bone formation and resorption; however, this change is transient. The full therapy is associated with suppression of bone turnover.

## 1. Introduction

Glucocorticoids (GCs) are widely used for various autoimmune diseases because of their immunosuppressive and anti-inflammatory effects. Intravenous methylprednisolone (IVMP) pulse therapy is currently considered as the first-line treatment for moderate-to-severe and active Graves’ orbitopathy (GO) according to the European Group on Graves’ Orbitopathy (EUGOGO) recommendations [1]. This therapy is associated with the risk of side effects including cardiac arrythmias, pulmonary embolism, elevation of liver enzymes, and suppression of the hypothalamic–pituitary–adrenal (HPA) axis [2,3,4,5,6].

In contrast to the well-known dose-dependent negative impact on bone tissue associated with the treatment with oral GCs, it remains unclear whether intravenous GCs exert the same effect. Patients receiving oral GCs even in small doses may have bone loss and increased risk of fractures [7]. According to the current evidence, there is no safe threshold dose below which oral GCs have no effect on bone tissue [8]. The adverse effects on bone are caused mainly by direct action of GCs on bone cells. In the initial phase, GCs rapidly reduce the differentiation of osteoblasts, increase the apoptosis of osteocytes and osteoblasts, increase the osteoclast generation, and prolong the lifespan of preexisting osteoclasts [9,10], which results in a decrease in bone formation and an enhancement of bone resorption. At the late stage, GCs decrease both osteoclast and osteoblast function [11], which leads to the low bone remodeling rate that characterizes chronic GC-induced bone disease [10]. Moreover, the oral GCs also indirectly affect bone adversely by suppressing sex steroid secretion [12], reducing calcium absorption in the gut [13], and renal tubular calcium reabsorption [14], thereby inducing the negative calcium balance.

The knowledge of above-mentioned deleterious effects of GCs on bone is based on the findings from studies involving patients treated with oral GCs, whereas the precise cellular basis of the influence of intravenous GCs remains elusive. Since intravenous GCs are given as an intermittent (pulse) therapy, often limited to a short period of time, there is a belief that their effect on bone is marginal [15,16,17]. There are studies, however, supporting the theory that such therapy is not so harmless in terms of bone safety [18,19,20]. A useful tool to assess the metabolic activity of the skeleton is the measurement of the bone turnover markers (BTM) which are released into the circulation during bone formation and resorption. The International Osteoporosis Foundation (IOF) in association with the International Federation of Clinical Chemistry and Laboratory Medicine (IFCC) has established that the reference analytes for bone formation and resorption markers are the N-amino terminal propeptide of type I procollagen (P1NP) and the C-terminal telopeptide of type I collagen (CTX), respectively [21]. The number of reports assessing the effects of intravenous GCs on BTM is limited. There are studies demonstrating the suppression of bone formation (expressed by a decrease in different BTM) occurring immediately after very high doses of intravenous GCs [19,20,22,23,24,25]; however, none has done so after 0.5 g of IVMP. The impact on bone resorption remains to be elucidated.

The aim of our study was to investigate the effects on BTM (P1NP and CTX) occurring at the very beginning of IVMP therapy as well as after the full course of treatment with 12 IVMP pulses given in a weekly schedule with a cumulative dose of 4.5 g in patients with GO. At the same time, we evaluated alterations in the calcium metabolism parameters. To the best of our knowledge, the acute impact of a single pulse of 0.5 g IVMP on bone metabolism has not been previously analyzed.

## 2. Materials and Methods

### 2.1. Patients

Twenty-three patients were consecutively recruited from the Department of Endocrinology, Medical University of Warsaw, between 2019 and 2021. Patients were eligible for inclusion in the study if they were over 18 years of age; were diagnosed with active, moderate-to-severe GO based on EUGOGO recommendations [1]; were euthyroid, with free triiodothyronine (fT3) and free thyroxine (fT4) levels within the reference range at least one month preceding, as well as throughout the study. Exclusion criteria were the following: treatment with GCs or any other treatment known to significantly alter bone metabolism (e.g., bisphosphonates or other drugs with anti-fracture effects, proton pomp inhibitors, selective serotonin reuptake inhibitors, benzodiazepines, antiepileptic, antipsychotic drugs, heparin, vitamin K antagonists) within the last 12 months; presence of the following co-morbidities: hyper- or hypoparathyroidism, rheumatoid arthritis, severe liver, or kidney disfunction; clinical diagnosis of osteoporosis based on the presence of low-energy fractures, or BMD measurements (DXA T score below −2.5 SD). The study was approved by the Local Bioethics Committee No. KB/225/2018 and was conducted in accordance with the Declaration of Helsinki. Written informed consent was obtained from all individual participants included in the study. The characteristics of the study group are presented in Table 1.

### 2.2. Study Design

All the patients were admitted to the hospital (on Day 0) and on the following day (on Day 1) received 0.5 g of IVMP over 4 h beginning at 8 a.m. Intravenous flow regulator (Exadrop B.Brown) was used in order to obtain infusion rates with precision. Afterwards, IVMP pulses were continued according to the EUGOGO guidelines: treatment duration was 12 weeks with once-weekly intravenous pulses (starting dose of 0.5 g once weekly for 6 weeks, followed by 0.25 g once weekly for 6 weeks with a cumulative dose of 4.5 g) [1]. Supplementation with 1.0 g of calcium and 2000 IU of vitamin D daily was routinely provided at the beginning of IVMP therapy and continued throughout the study. Seven patients were excluded in the course of the study (due to increased level of aspartate and alanine aminotransferases in 3 patients, thyroidectomy during IVMP therapy in 2 patients, and suspension of the treatment due to COVID-19 pandemic in 2 patients). The rapid changes in bone metabolism occurring over days following first IVMP pulse were evaluated in a subset of 16 patients.

### 2.3. Biochemical Evaluation

Blood samples were collected on Day 0 (at 4 p.m). On Day 1, the first IVMP pulse was launched at 8 a.m. Serum and 2 h urine samples were obtained on Day 1 (before the IVMP pulse at 8 a.m. and then, four times every 2 h after the beginning of IVMP infusion), once on Day 2 and then, one week after the first IVMP pulse (on Day 8, once at 8 a.m.). Finally, the blood samples were collected after 12 weeks of IVMP therapy (at 8 a.m.). A scheme of the study design with blood and urine sampling time-points in early (within first days) and further (after the IVMP therapy) evaluation is presented in Figure 1.

Bone turnover markers (P1NP and CTX) were evaluated on Day 0 (at 4 p.m.), on Day 1 (twice: at 8 a.m. and then 8 h after the beginning of the first IVMP pulse), and on Days 2 and 8 (once, at 8 a.m.). The serum samples were obtained on Day 0 (before IVMP therapy) to avoid the effect of the circadian rhythm on the levels of BTM. The results were compared with the corresponding time points (4 p.m. on Day 1 with 4 p.m. on Day 0; 8 a.m. on Days 2 and 8 with 8 a.m. on Day 1). P1NP and CTX were also assessed after the full course of IVMP pulse therapy (4.5 g IVMP divided into 12 once-weekly pulses).

The following calcium metabolism parameters (Ca-P) were measured on Day 1 (five times), on Day 2 (once), and on Day 8 (once): serum levels of Ca, P, iPTH, creatinine (Cr) as well as Ca, P, and Cr in urine. Afterwards, the fractional excretions of calcium (FeCa = urine Ca × serum Cr/serum Ca × urine Cr), as well as tubular maximum reabsorption of phosphate adjusted to GFR (TmP/GFR = serum P − (urine P × serum Cr/urine Cr)) were calculated.

Vitamin D metabolites (vit-D): 1,25-dihydroxyvitaminD (1,25(OH)_2_D) and 25-hydroxyvitamin D (25(OH)D) were measured on Day 1 (at 8 a.m. and 4 p.m.) and on Days 2 and 8 (at 8 a.m.).

One week after the full course of IVMP therapy, 25(OH)D as well as iPTH, were assessed.

The 25-hydroxyvitamin D concentrations below 20 ng/mL were described as deficient, concentrations of 20–30 ng/mL as suboptimal, and concentrations higher than 30 ng/mL as optimal vitamin D status according to the guidelines for vitamin D supplementation approved in Central Europe [26].

We measured Cr, iPTH, 25(OH)D, TSH, fT4, fT3, and thyrotropin receptor antibodies (TRAB) using an electrochemiluminescence immunoassay on Cobas 8000 Analyzer (Roche Diagnostics, Mannheim, Germany). Ca and P were analyzed calorimetrically. The concentrations of 1,25(OH)_2_D, P1NP, and CTX (CrossLaps) were assessed using automated chemiluminescence immunoassay (CLIA) by the Immunodiagnostic system IDS iSYS. 

The normal ranges were as follows: Ca, 2.15–2.6 mmol/L; P, 0.81–1.45 mmol/L; Cr, 0.5–1.0 mg/dl; iPTH, 15–65 pg/mL; TSH, 0.27–4.2 µIU/mL; fT3, 3.1–6.8 pmol/L; fT4, 12.0–22.0 pmol/L, P1NP, 27.7–127.6 ng/mL; CTX, 0.038–0.724 ng/mL in men, 0.034–0.635 ng/mL in premenopausal women, 0.034–1.037 ng/mL in postmenopausal women; 1,25(OH)_2_D, 15.2–90.1 pg/mL

### 2.4. BMD Evaluation

We evaluated BMD of the lumbar spine (L1–L4) and the femoral neck at baseline using Hologic Discovery A Densitometer and all the scans were analyzed using software version 12.6 (Bedford, MA, USA). All BMD measurements performed up to 4 weeks prior to the first IVMP pulse were considered the baseline results and subsequently analyzed by the same physician. In postmenopausal women and men over 50, osteoporosis and osteopenia were diagnosed with a T-score of the lumbar spine and/or the femoral neck ≤ −2.5 standard deviation (SD) and between <−1.0 and >−2.5 SD, respectively.

### 2.5. Statistical Methods

All analyses were performed using SPSS statistical software version 22.0 (IBM SPPS Statistics, New York, NY, USA). Continuous variables are expressed as means ± SD, while categorical variables are expressed as numbers (*n*) and percentages (%). The Shapiro–Wilk test was used to confirm or reject the normal distribution of each continuous variable. Comparisons between continuous data were performed using paired *t*-test (for parameters with normal distribution) or Wilcoxon rank sum test (for parameters with distribution deviations). Pearson correlation test was performed to investigate correlations. Statistical significance was established for results with *p* value < 0.05.

## 3. Results

### 3.1. Markers of Bone Turnover and Calcium Metabolism Parameters at Baseline

At baseline, there were positive correlations between P1NP and CTX (*r* = 0.54, *p* value = 0.008).

No significant correlations were observed between BTM and age, BMI, TRAB, TSH, fT3, 25(OH)D, or lumbar spine bone mineral density (BMD) or femoral neck BMD. P1NP correlated negatively with serum fT4 (*r* = −0.49, *p* value = 0.02). Details are presented in Appendix A.

Pretreatment biochemical indices of Ca, P, iPTH, fT3, and fT4 were within the normal range. There was no correlation between basal 25(OH)D and iPTH levels.

### 3.2. Early Effects of a Single Dose of 0.5 G IVMP on BTM

There was a rapid fall in P1NP already evident 8 h after the IVMP pulse (reaching 66% of the baseline value) (Figure 2 and Figure 3). Twenty-four hours after the IVMP dose, serum P1NP levels further declined (to 41% of the baseline). Then, one week later, P1NP level was not significantly different than before IVMP therapy.

We observed a reduction of CTX level occurring 7 days after the starting of the first IVMP pulse (to 72% of the baseline value) (Figure 2 and Figure 3).

We found a positive correlation between P1NP and CTX (*r* = 0.50, *p* value = 0.049) one week after the first IVMP pulse.

### 3.3. The Influence of the Full Course of IVMP Pulse Therapy on BTM and Calcium Metabolism Parameters

P1NP and CTX decreased significantly after the full course of IVMP treatment reaching 83% and 71% of the baseline value, respectively (Figure 3 and Figure 4, Table 2). We found a positive correlation between P1NP and CTX after 12 weeks (*r* = 0.91, *p* value < 0.001). No change of 25(OH)D or iPTH was noticed after IVMP therapy.

### 3.4. Early Effects of a Single Dose of 0.5 G IVMP on Calcium Metabolism Parameters and 1,25(OH)_2_D

Serum phosphate (P) fell 2 h after the start of IVMP administration, which was associated with an immediate drop in tubular maximum reabsorption of phosphate adjusted to GFR (TmP/GFR) (Figure 5). Serum P and TmP/GFR were back to normal on the next day.

Serum calcium (Ca) decreased within 8 h after the IVMP administration, which was accompanied by an increase of fractional excretion of calcium (FeCa). On the following day, we observed a significant increase in Ca associated with a decrease in FeCa. Throughout the course of IVMP administration, there were no changes in ionized calcium.

Serum 1,25(OH)_2_D increased rapidly, peaked 24 h after IVMP administration, and returned to baseline within 7 days (Figure 6). Throughout the course of IVMP administration, there were no changes in 25(OH)D.

Changes in serum Ca and P were all independent of iPTH, which did not change during this time.

## 4. Discussion

In the current study, we prospectively investigated the influence of IVMP pulse therapy on BTM and calcium metabolism parameters in patients with GO. We demonstrated that a single pulse of 0.5 g of IVMP reduced both P1NP and CTX concentrations. The decrease of bone formation marker was rapid but transient, as P1NP was back to normal after one week. The reduction of bone resorption was delayed, as the fall in CTX was found after one week following IVMP pulse. When IVMP pulses were repeated once weekly for 3 months with a cumulative dose of 4.5 g, the reduction of P1NP and CTX was found again with evaluation carried out one week after the IVMP pulses.

The measurement of P1NP provides information on the rate of the synthesis of type I collagen by osteoblasts during osteogenesis [27]. Therefore, the decreased level of P1NP is indirect evidence of suppressed bone formation [7]. The findings of our research confirmed the observation that the therapy with intravenous GCs is associated with the inhibition of bone formation, which is considered to be a cardinal feature of the oral GCs impact on bone tissue [14]. The design of our study allowed us to show that the suppressive effect of intravenous GCs on osteoblastic activity is evident already after an intermediate dose of 0.5 g of IVMP, but the phenomenon is transient, which stays in agreement with previous studies in which higher doses of IVMP were used [19,23]. However, the decrease of P1NP persistent after the full course of IVMP therapy in our study indicates that the cumulative effect of intravenous GCs on bone may be dose-dependent, similarly to the oral GCs [12]. Chen et al. evaluated the effect of various IVMP therapy regimens in GO on BTM (using the same cumulative dose of 4.5 g over 12 weeks versus over 4 weeks), showing similar decrease in P1NP [25]. Hu et al. showed the decrease in P1NP only after the second cycle of IVMP therapy with a cumulative dose of 4.5 g [15]. The question arises whether there is a threshold above which intravenous GCs exert negative effect on bone tissue.

Based on the findings of our study, bone resorption appears to be suppressed by IVMP therapy. CTX is a telopeptide derived from the degradation of type I collagen with enzymes produced by osteoclasts during bone resorption [27]. The decrease of this sensitive marker is evidence of suppressed bone degradation [7]. CTX is characterized by large circadian variations with the lowest concentration in the afternoon [27]. Therefore, we compared the levels with the corresponding hour of the preceding day. In our study, CTX levels were correlated to P1NP levels when the analysis was made at baseline, and between P1NP and CTX after one week as well as after the whole course of IVMP therapy. Intravenous GCs probably exert a primary effect on cells of the osteoblastic lineage and the reduction of osteoblast number may result in decreased stimulation of osteoclastogenesis and decreased bone resorption, similarly as it occurs during the long-term treatment with oral GCs [14]. Data regarding the resorptive effect of GCs are conflicting [28]. Previous studies suggest that excessive bone resorption occurs especially in the early phase during oral GCs therapy [14,29]. In reports assessing the impact of intravenous GCs, bone resorption was decreased [15,16,18,25] or increased [19,20], depending on the time of measurement and the parameter analyzed (Table 3). However, it should be noted that in all studies involving GO patients treated with repeated IVMP pulses, a decrease of bone resorption markers was found [15,16,18,25]. It seems that IVMP therapy in GO does not increase bone resorption but may result in its inhibition.

The changes in P1NP and CTX are particularly significant given the shifts in calcium metabolism parameters observed in our study early after the beginning of IVMP administration. No change of iPTH levels during IVMP therapy implies that secondary hyperparathyroidism does not play a role in bone alterations induced by intravenous GCs, which is consistent with previous studies [20,30]. GCs diminish duodenal calcium absorption [13], exert direct effect on renal phosphate handling [31], and increase renal Ca wasting through overstimulation of non-selective mineralocorticoid receptor [32]. In our research, we observed a rapid decrease in serum P and Ca levels following IVMP pulse, which was accompanied with the abrupt reduction of renal P reabsorption and increase of renal Ca wasting. These changes preceded a subsequent increase of serum 1,25(OH)_2_D. Active vitamin D is a potent agent increasing Ca and P absorption from intestine and decreasing renal Ca and P losses [33]. There is a possibility that the brief rise of 1,25(OH)_2_D after IVMP was responsible for the normalization of the P level, and a significant increase in serum Ca noted on the following day of our research. The latter observation is consistent with our previous study [34].

There are at least three coexisting independent factors other than IVMP that influence bone metabolism in patients with GO and should be discussed: former hyperthyroidism presents in the majority of the study group, the inflammatory state as GO is an autoimmune disease, and vitamin D status. Firstly, it is known that thyrotoxicosis enhances bone turnover with predominant influence on bone degradation, leading to gradual decrease in BMD [35]. After attainment of euthyreosis, bone resorption rapidly ameliorates, whereas bone formation remains stimulated for at least several months [36] and is followed by an increase in BMD. In our study, all of our patients were euthyroid at least one month prior to the IVMP therapy as well as during the study. At baseline, we found a negative correlation between P1NP and fT4 level. It cannot be excluded that in patients with hyperthyroidism in the past, the duration of euthyroidism before the start of IVMP therapy might have been too short for normalization of bone turnover. However, the reduction of bone turnover observed after the whole IVMP pulse therapy indicates that the ongoing bone remodeling might have been disrupted by IVMP. Secondly, patients with GO have an increased release of pro-inflammatory cytokines into the circulation, such as TNFα, IL-6, and IL-1, which stimulate osteoclast activity [37]. There is a possibility that the decrease of bone resorption observed after IVMP in our study may be secondary to a suppression of inflammatory process influencing the rate of bone loss. Finally, nearly optimal baseline levels of vitamin D in our study group and sufficient ongoing vitamin D supplementation ensured no additional negative effect on bone turnover.

Taking everything into consideration, the question arises whether the significant disturbances in bone metabolism and suppressed bone turnover observed after IVMP therapy in the current study exert long-term negative effects on bone tissue. Although the reduction of bone turnover following IVMP pulse therapy might be a marker for dysfunctional bone remodeling, our results should be interpreted with caution until direct evidence for harmful influence of intravenous GCs on bone such as increased risk for fractures has been found. So far, such studies evaluating fracture prevalence in patients treated with intravenous GCs have not been conducted.

Our study has certain limitations that should be acknowledged. The main limitation is a relatively small sample size including men and pre- and postmenopausal women. Moreover, the reevaluation of BTM was made after a short follow-up period. The design of this research allowed us to show that IVMP pulse therapy is associated with significant alterations in bone metabolism. However, the data regarding the long-term effect of IVMP on BTM and vitamin D are limited. The question arises whether the changes observed in our study are reversable after withdrawing GCs. We have to bear in mind that in case of patients with GO, if response to primary treatment is poor, the second-line treatment is considered, including second course of IVMP with higher cumulative dose (7.5 g), or oral GCs, which further affect bone metabolism. Finally, the predictive value of BTM is limited by their large biological variation (influence of circadian rhythm, age, food intake, kidney function, and pregnancy status) [27]. Nevertheless, BTM has clinical utility and bring additional value to BMD as changes in BTM often occur before improvement or loss of BMD can be detected. Future studies with longer follow-up period should include analysis of DXA parameters, trabecular bone score, bone turnover markers and active research for fracture prevalence.

## 5. Conclusions

Bone formation as well as bone resorption are suppressed after IVMP given in weekly pulses with a cumulative dose of 4.5 g in patients with GO.

An intermediate dose of 0.5 g of IVMP causes rapid, but transient decrease in bone formation, followed by subsequent inhibition of bone resorption.

## Figures and Tables

**Figure 1 jcm-11-05005-f001:**
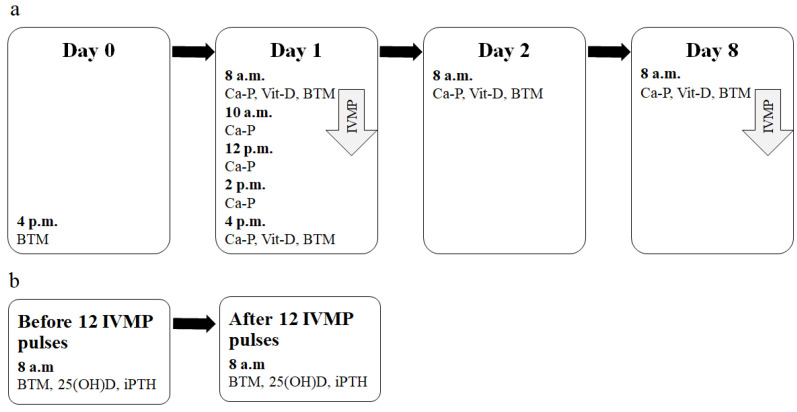
The scheme of the study design. (**a**) Early evaluation time-points and parameters measured (after a single IVMP pulse). (**b**) Further evaluation time-points and parameters measured (after the full IVMP therapy). Ca-P—Evaluation of calcium metabolism parameters (Ca, P and Cr in serum and in urine, serum iPTH). Vit-D—Evaluation of vitamin D metabolites (25(OH)D and 1,25(OH)_2_D); BTM—Evaluation of BTM (P1NP, CTX). 1,25(OH)_2_D, 1,25-dihydroxyvitaminD; CTX, C-terminal telopeptide of type I collagen; BTM, bone turnover markers; IVMP, intravenous methylprednisolone; 25(OH)D, 25-hydroxyvitaminD, iPTH, intact parathormone; P1NP, amino-terminal propeptide of type I procollagen.

**Figure 2 jcm-11-05005-f002:**
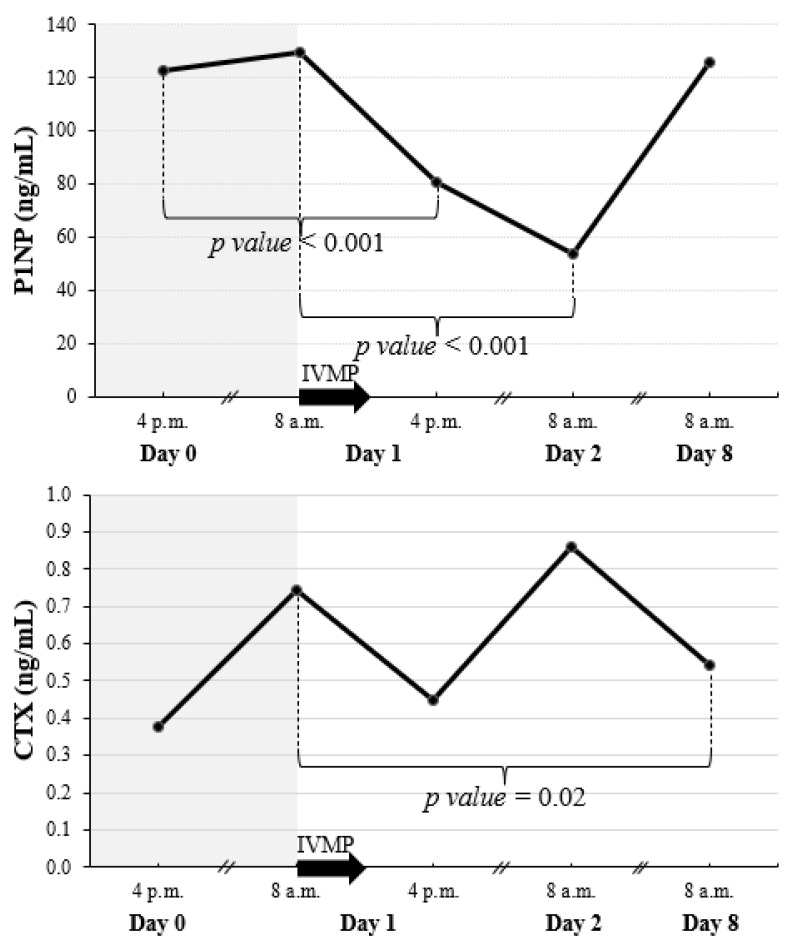
Early changes of P1NP and CTX after a single pulse of 0.5 g IVMP. The black arrow represents IVMP pulse of 0.5 g. When significant difference between consecutive values was found, *p* value is given. CTX, C-terminal telopeptide of type I collagen; IVMP, intravenous methylprednisolone; P1NP, amino-terminal propeptide of type I procollagen.

**Figure 3 jcm-11-05005-f003:**
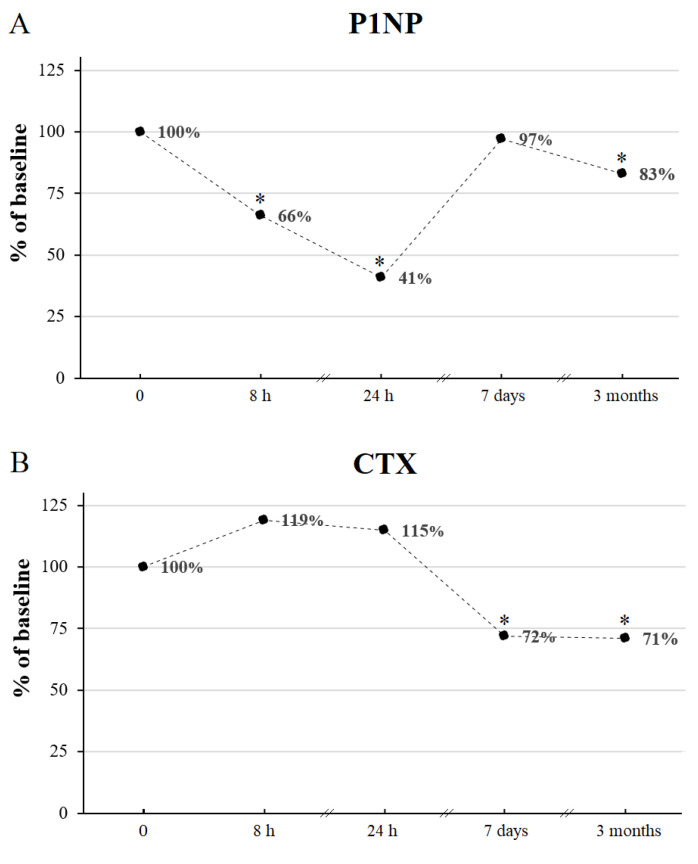
The percentage changes of P1NP (**A**) and CTX (**B**) during IVMP therapy with respect to the corresponding baseline values (0). * *p* value < 0.05 vs. corresponding time point before IVMP therapy. CTX, C-terminal telopeptide of type I collagen; h, hours IVMP, intravenous methylprednisolone; P1NP, amino-terminal propeptide of type I procollagen.

**Figure 4 jcm-11-05005-f004:**
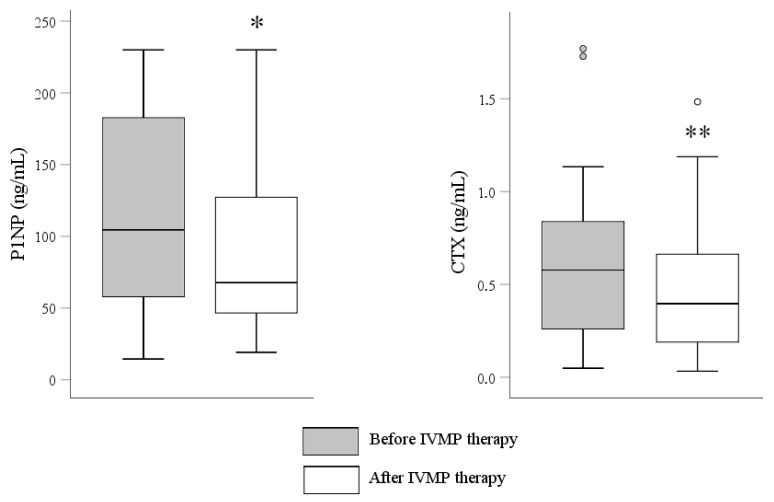
Effect of IVMP therapy (12 weeks with once-weekly intravenous pulses with a cumulative dose of 4.5 g) on P1NP and CTX. * *p* value < 0.05 vs. before IVMP therapy; ** *p* value < 0.01 vs. before IVMP therapy; CTX, C-terminal telopeptide of type I collagen; IVMP, intravenous methylprednisolone; P1NP, amino-terminal propeptide of type I procollagen.

**Figure 5 jcm-11-05005-f005:**
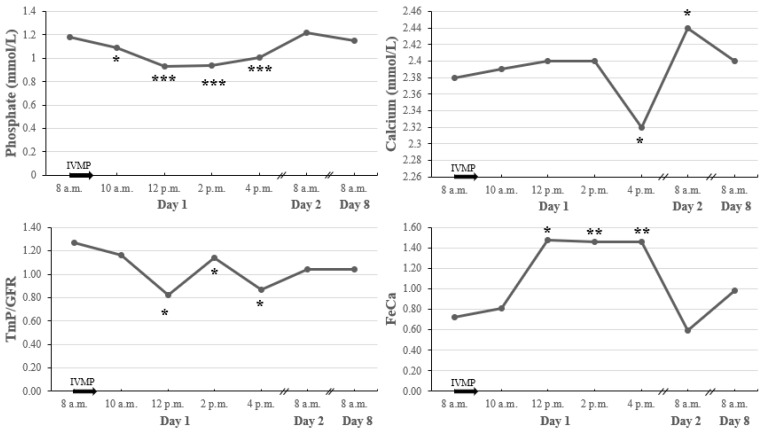
Early changes in calcium metabolism parameters after a single pulse of 0.5 g IVMP. * *p* value < 0.05 vs. baseline; ** *p* value < 0.01 vs. baseline; *** *p* value < 0.001 vs. baseline. The black arrow represents IVMP pulse of 0.5 g; IVMP intravenous methylprednisolone; TmP/GFR, tubular maximum reabsorption of phosphate adjusted to GFR; FeCa, fractional excretion of calcium.

**Figure 6 jcm-11-05005-f006:**
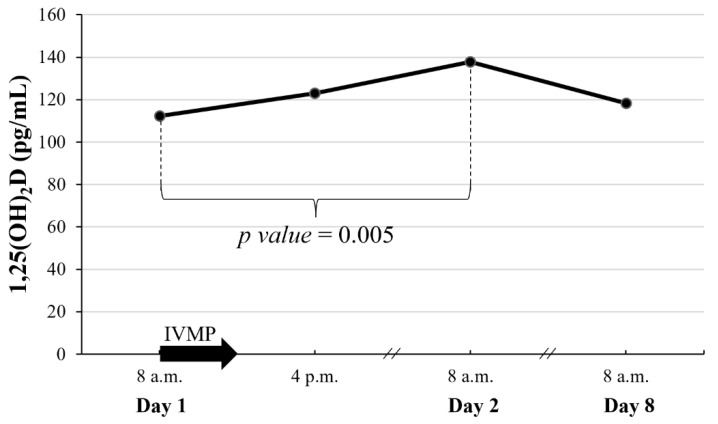
Early changes of 1,25(OH)_2_D after a single pulse of 0.5 g IVMP. The black arrow represents IVMP pulse of 0.5 g. When significant difference between consecutive values was found, *p* value is given; 1,25(OH)_2_D, 1,25-dihydroxyvitaminD; IVMP, intravenous methylprednisolone.

**Table 1 jcm-11-05005-t001:** Baseline characteristics of patients (*n* = 23).

Variable	Number of Patients (%) or Mean ± SD (Range)
Age (years)	54 ± 12 (31 ÷ 74)
Female sex	20 (87%)
Postmenopausal women	11 (55%)
Body mass index (kg/m^2^)	26.5 ± 6.2 (17.1 ÷ 41.3)
Thyroid disease	
Graves’ disease treated for hyperthyroidism	15 (65%)
Graves’ disease after radical treatment	6 (26%)
Hashimoto thyroiditis on levothyroxine	2 (9%)
Duration of treatment with antithyroid drugs before IVMP therapy ^1^ (month)	10.0 ± 5.0 (3 ÷ 18)
TSH (normal range: 0.27–4.2 µIU/mL)	1.5 ± 1.5 (0.005 ÷ 3.9)
fT4 (normal range: 12.0–22.0 pmol/L)	16.2 ± 4.2 (12.2 ÷ 22.0)
fT3 (normal range: 3.1–6.8 pmol/L)	4.6 ± 0.9 (3.1 ÷ 6.6)
TRAB (normal range: <1.75 IU/L)	17.6 ± 12.4 (2.8 ÷ 40)
25(OH)D	28.0 ± 9.5 (7.2 ÷ 57.0)
Deficiency	3 (13%)
Suboptimal	9 (39%)
Optimal	11 (48%)
P1NP (normal range: 27.7–127.6 ng/mL)	121.2 ± 67.02 (14.4 ÷ 230.0)
CTX (normal range: 0.038–0.724 ng/mL in men, 0.034–0.635 ng/mL in premenopausal women, 0.034–1.037 ng/mL in postmenopausal women)	0.7 ± 0.5 (0.05 ÷ 1.8)
iPTH (normal range: 15–65 pg/mL)	44.9 ± 11.2 (23 ÷ 62.2)
Lumbar BMD at baseline	1.0 ± 0.1 (0.8 ÷ 1.3)
Normal body mass	16 (70%)
Osteopenic	7 (30%)
Femoral neck BMD at baseline	0.7 ± 0.5 (−1.0 ÷ 1.4)
Normal body mass	16 (70%)
Osteopenic	7 (30%)

^1^ Duration of treatment with antithyroid drugs before IVMP therapy is presented for 15 patients with Graves’ disease treated for hyperthyroidism. 25(OH)D, 25-hydroxyvitamin D; BMD, bone mineral density; CTX, C-terminal telopeptide of type I collagen; fT4, thyroxine; fT3, triiodothyronine; SD, standard deviation; P1NP, amino-terminal propeptide of type I procollagen; TBS, trabecular bone score; TSH, thyroid stimulating hormone; TRAB, thyrotropin receptor antibodies.

**Table 2 jcm-11-05005-t002:** Parameters of bone turnover before and after 12 IVMP pulses given in a weekly schedule with a cumulative dose of 4.5 g in 16 patients with Graves’ orbitopathy.

Variable	before IVMP	after IVMP	*p* Value
P1NP (ng/mL)	114.6 ± 70.8	95.38 ± 70.85	0.04
CTX (pg/mL)	0.7 ± 0.5	0.5 ± 0.4	0.002
25(OH)D (ng/mL)	29.9 ± 9.9	30.9 ± 8.3	0.49
iPTH (pg/mL)	45.1 ± 11.5	41.7 ± 6.4	0.18

CTX, C-terminal telopeptide of type I collagen; IVMP, intravenous methylprednisolone; iPTH, intact parathormone; P1NP, amino-terminal propeptide of type I procollagen.

**Table 3 jcm-11-05005-t003:** Summary of studies that investigated acute changes in levels of serum bone markers in patients treated with IVMP.

Study	Size of the Group (*n*)	Diagnosis	IVMP Regimen	BTM Evaluation	Results—Markers of Bone Formation	Results—Markers of Bone Resorption
Cosman [20] 1994	56	MS	1.0 g over 1 h daily for 10 days, 0.5 g/day for 2 days, 0.25 g/day for 2 days (cumulative dose 11.5 g)	In 9 pts: at baseline, after 1, 3, 5, 8 h, and then after 3 days and 2 weeks	Decrease of OC after 8 h persistent though the IVMP therapy	Increase of TRAP after 2 weeks
Peretz [23] 1996	7	RD	1.0 g over 30 min	At baseline and after 3, 6, 12, 24, 48, 72 h	Decrease of OC after 6, 24 h, Decrease of P1CP after 12 h	
Ardissone [24] 2002	23	MS	1.0 g daily for 10 days (cumulative dose 10.0 g)	At baseline, and after 3, 10, 30 days	Decrease of OC after 3 and 10	
Dovio [19] 2004	13	MS	15 mg/kg daily for 10 days	At baseline and every day over 10 days, and then after 3 months	Decrease of OC and P1NP after 48 h, increase of BTM after 3 months	Increase of CTX after 1 week, and 3 months
Gasińska [18] 2012	30	GO	1.0 g twice a week over 4 weeks (cumulative dose 8.0 g)	At baseline and after 1 month	Decrease of OC and P1CP after 1 month	Decrease of ICTP after 1 month
Censi [16] 2018	11	GO	In 6/11 pts: 0.5 g/week for 6 weeks, then 0.25 g/week for 6 weeks (cumulative dose 4.5 g) In 4/11 pts—cumulative dose—1.5 g; in 1 patient—5.25 g 300,000 IU of vitamin D3 a week before IVMP	At baseline, after 1 week, and then 1, 3, 6 and 12 months	No change of P1NP	Decrease of CTX after 1 month
Hu [15] 2020	45	GO	0.5 g/week for 6 weeks, then 0.25 g/week for 6 weeks (cumulative dose 4.5 g) In a subset of 16 pts—the 2nd course of IVMP pulse therapy 0.25 ug of alfacalcidol and 600 mg of calcium daily	At baseline and after 3 months In a subset of 16 pts—3 months after the start of the 2nd course	Decrease of P1NP after 3 months in a subset of 16 pts subjected to a 2nd course of IVMP therapy	Decrease of CTX after 3 months
Chen [25] 2021	48	GO	In 26/48 pts: 0.5 g/week for 6 weeks, then 0.25 g/week for 6 weeks (cumulative dose 4.5 g) In 22/48 pts: 0.5 g for three consecutive days/twice every two weeks, then 0.25 g for 3 consecutive days/twice every two weeks	At baseline and after 3 months	Decrease of P1NP Decrease of P1NP	Decrease of CTX Decrease of CTX
Current study	23	GO	0.5 g/week for 6 weeks, then 0.25 g/week for 6 weeks (cumulative dose of 4.5 g) 2000 IU of vitamin D3 and 1.0 g of calcium daily	At baseline, after 8, 24 h, and then after 1 week and 3 months	Decrease of P1NP after 8 and 24 h, normalization after 1 week, and then decrease after 3 months	Decrease of CTX after 1 week, and after 3 months

IVMP, intravenous methylprednisolone; GO, Graves’ disease; MS multiple sclerosis; RD, various rheumatic dise ases; BTM, bone turover markers; pts, patients; P1NP, amino-terminal propeptide of type I procollagen; CTX, C-terminal telopeptide of type I collagen; OC, osteocalcin; P1CP, procollagen type 1 carboxyterminal propeptide; ICTP, cross-linked carboxyterminal telopeptide of type I collagen; TRAP, tartrate-resistant acid phosphatase.

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
