# Peer review of "Decrease in Bone Formation and Bone Resorption during Intravenous Methylprednisolone Pulse Therapy in Patients with Graves’ Orbitopathy"

_jcm, 2022, doi:10.3390/jcm11175005_

Round 1
Reviewer 1 Report
The manuscript titled “Decrease in Bone Formation and Bone Resorption during Intra-venous Methylprednisolone Pulse Therapy in Patients with Graves’ Orbitopathy” evaluated short term effect of IV steroid therapy on changes in bone turnover markers (BTM; PINP, CTX). This prospective study has provided valuable data that is difficult to obtain. There are a few points to be addressed for the better result.
1. When was the last blood sampling done after 12 weeks of IV steroid treatment? It was stated that PINP and CTX concentrations decreased 8 hours after the first IV steroid. Did the last measurement decrease more than the decrease after first 8 hours? I would like you to add the last PINP and CTX concentrations measured at 12 weeks to fig2 or show the data in comparison with the initial changes.
2. Please indicate statistical significance in Figure 3
3. What is fosforany in Figure 4?
4. In PINP, terms procollagen and collagen are used interchangeably in the paper. Please correct it.
5. This paper presents only short-term changes in BTM and vitamin D, and information on long-term effects is limited. Please further state the limitations of this study in ’Discussion’.
Author Response
Dear Sir/Madam,
We highly appreciate the time you have taken in reviewing this manuscript. We have found your comments and suggestions of great interest. It allowed us to improve our manuscript. We hope that after the responses and modification of the manuscript the final version would meet all the criteria for further process of edition.
Please see the attachment.
Yours sincerely,
Piotr Miśkiewicz

Reviewer 2 Report
General comments
In this manuscript, the authors investigated BTMs changes during intravenous methylprednisolone pulse therapy in patients with graves’ orbitopathy.
As you mentioned in the article, the sample size is too small, and diverse patients were included.
CTX has diurnal variability, you’d better to compare them with the same time points.
The authors should comment variability of CTX explain how pulmonary dysfunction could affect SIs independently of BMD and/or why SIs could be better than BMD as a clinical parameter.
Specific comments
1) in Table 1, please comment unit on the duration of treatment (maybe a month?)
2) in results
The authors described that CTX increased at 8 hours after starting the first IVMP pulse. Please check that. Also, I am not sure that Its decreases meant a circadian rhythm.
Author Response

(The authors gave the same response as above.)
